

# On the quasi-steady vorticity balance in the mature stage of hurricane *Irma (2017)*

Jasper de Jong[1], Michiel L. J. Baatsen[1], and Aarnout J. van Delden[1]

[1]Institute for Marine and Atmospheric research Utrecht, Princetonplein 5, 3584 CC Utrecht, The Netherlands

**Correspondence:** Jasper de Jong (j.dejong3@uu.nl)

**Abstract.** Tropical cyclone (*TC*) intensification is a process depending on many factors related to the thermodynamical state and environmental influences. It remains a challenge to accurately model *TC* intensity due to the role of unsteady features like deep convective bursts, boundary layer dynamics and eddy processes. The impermeability theorem for potential vorticity substance, *PVS*, on isentropic surfaces provides a way to analyze the absolute vorticity structure and tendency in *TC*s. We will examine this theorem in a numerical simulation of hurricane *Irma (2017)* near lifetime-peak intensity. Hurricane *Irma* was a very intense hurricane that persisted as a category five hurricane on the Saffir-Simpson intensity scale for three consecutive days, the longest for any Atlantic hurricane since satellite observations. During this period the intensity of *Irma* was remarkably constant. According to the impermeability theorem, the radially outward vorticity flux due to divergence above the atmospheric boundary layer must be compensated by an equally strong radially inward vorticity flux due to the effect of diabatic heating in the presence of vertical wind shear. The model results agree with this theorem and we find a strong anticorrelation between the advective and diabatic components of the radial vorticity flux. The impact of parametrized turbulence on the vorticity balance is found to be weak and does not explain the residual flux that would otherwise close the vorticity balance.

## 1 Introduction

Despite the organized appearance of tropical cyclones (*TCs*), their predictability remains an active research topic. Track forecasts have improved steadily during the past half century and Landsea and Cangialosi (2018) suggest that we might be near its limit of predictability. Intensity forecasts however, have seen modest improvements until a decade ago (Cangialosi et al., 2020). As *TC*s are known to cause widespread damage and fatalities among populations, it remains important to enhance our understanding of the intensity of these complex weather systems. This study aims to contribute to our understanding of *TC* intensity by analyzing the budget of vorticity in the mature, quasi-steady stage of hurricane *Irma* (2017).

Dry Rossby-Ertel Potential Vorticity, $P$, is a measure of the rotation of a column of fluid. It relates to isentropic absolute vorticity, $\eta = f + \zeta$, where $f$ is the coriolis parameter and $\zeta = \frac{\partial v}{\partial x} - \frac{\partial u}{\partial y}$ isentropic relative vorticity, with $u$ and $v$ ($x$ and $y$) representing the eastward and northward velocities (distance), respectively, by $P = \eta\sigma^{-1}$, where $\sigma$ is the mass per unit $xy\theta$-





space and $\theta$ denotes potential temperature. $P$ is an essential quantity for tracking most synoptic features due to its conservation
under adiabatic conditions. By the invertibility principle, knowing the distribution of $P$ enables the wind field to be determined
diagnostically assuming hydrostatic and gradient wind balance (Hoskins et al., 1985). Estimates of $P$ in *TCs* derived from
reconnaissance aircraft measurements may reveal a hollow vorticity tower structure (Martinez et al., 2019), most commonly for
intensifying *TCs* (Rogers et al., 2013). These towers are also seen in numerical models models. Progress in intensity forecast
accuracy during the recent decade is attributed to the improvement of numerical weather prediction (*NWP*) models, model
consensus aids, forecaster guidance and classification of rapid intensification predictors, nowadays supported by machine
learning techniques (Cangialosi et al., 2020).

The evolution of this dry-adiabatic $P$ can be determined from horizontal advection as there is no cross-isentropic mass flow.
The tendency of $\eta$ then takes the form:

$$\frac{d}{dt}\int_V \eta\, dV = -\oiint_A \boldsymbol{J}_{dry} \cdot d\boldsymbol{A} \tag{1}$$

where $\boldsymbol{J}_{dry} = \boldsymbol{v}_h \eta$ represents the adiabatic vorticity flux vector, $\boldsymbol{v}_h$ denotes the isentropic wind vector and we integrate over
volume $V$ in $xy\theta$-space, having enclosing area $A$. As the air in a *TC* above the boundary layer generally diverges from the
centre, this would lead to a negative tendency of $\eta$, i.e. a spin down of the vortex. Clearly the adiabatic approximation is not
sufficient in a *TC* environment.

Additional vorticity flux components are responsible for maintaining a steady state vortex. In this paper, we evaluate the
diabatic heating component of the vorticity flux as given by Haynes and Mcintyre (1987), add a turbulence component and
examine whether the change in flux is strong enough to cancel out the divergence due to advection, whether temporal changes
in the advective vorticity flux convergence due to transverse circulation changes leave their imprint in the remaining flux
components, whether the vorticity budget is closed and if parametrized turbulent transport might have a significant impact on
the total budget.

To answer these questions, numerical weather prediction (*NWP*) model data from hurricane *Irma* during its most intense
phase have been used. During the period of simulation, the intensity of *Irma* was notably steady. In fig. 1a we see the presence
of a hollow vorticity tower with maximum located on the inside of the eyewall, a feature mostly observed in intensifying *TCs*
(Rogers et al., 2013). Contours of radial velocity, $v_r$, confirm that the air on average diverges from the centre, especially in
the eyewall and outflow region. Isentropic surfaces indicate the extent of the warm core and a region of low static stability
from 360–370 K in the eye region, likely due to radiative cooling. Fig. 1b shows the lower troposphere mean absolute vorticity
along a zonal cross section as the simulation progresses. We may recognize an initial start up time of four hours during which
the monopole vorticity structure at the start of the simulation transforms into a cylindrical shape that sustains throughout the
remainder of the run. The intensity of the *TC* is considered stable enough to serve as an example of a steady-state mature
vortex. During the simulation, the radius of maximum vorticity fluctuates with a typical period of 4-6 hours, presumably due
to vortex Rossby waves (Wang, 2002). It however remains constant over the whole simulation, indicating the maturity of the
cyclone.





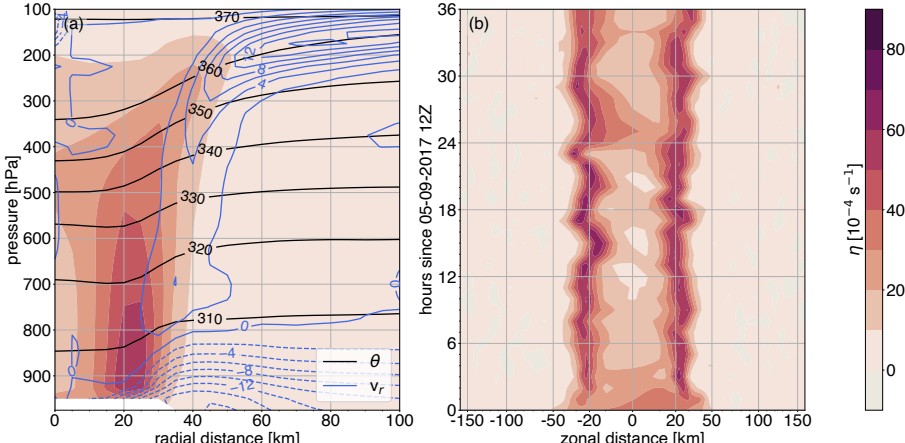

**Figure 1.** Azimuthal mean and lower-tropospheric mean zonal cross section of absolute vorticity. The data in panel **a** is time averaged over the 6/9 00:00 – 12:00Z period. Contour lines show potential temperature and radial wind with respect to the moving *TC* centre. Panel **b** contains the evolution of 500-900 hPa mean absolute vorticity along a zonal cross section through the centre.

The high resolution of *NWP* models allows the presence of hollow vorticity towers in simulations (e.g. Shuang et al., 2015; Tsujino and Kuo, 2020), instead of the more monolithic structure seen in coarser-scale models. These hollow vorticity towers are a feature observed in *TCs* (Martinez et al., 2019), and are directly related to their intensity. Their evolution is linked to
the amount of diabatic heating, which occurs in localized rotating deep convective bursts, also known as vortical hot towers (Hendricks et al., 2004). The intensity of these updrafts depends on convergence in the unsteady atmospheric boundary layer (Montgomery and Smith, 2017), where horizontal turbulent momentum diffusion is important (Rotunno and Bryan, 2012). Early pioneering work on modelling *TC* intensification often utilized relatively simple azimuthal mean models with a few vertical layers (Ooyama, 1969). These models are comprehensive and provide usefull insight, but they have intrinsic limitations
for understanding *TC* intensification (Persing et al., 2013). Recent efforts in *TC* modelling are therefore making frequent use of high resolution, convection-permitting, *NWP* models (Hazelton et al., 2020; Zhu et al., 2021). These efforts, recognizing the three-dimensional unsteady structure of a *TC*, would fall under the rotating convection paradigm coined by Montgomery and Smith (2017). By investigating an azimuthally averaged view of this paradigm, the comprehensiveness of azimuthal-mean models is combined with the ability of *NWP* models to resolve non-axisymmetric dynamics.
In early September 2017, hurricane *Irma* approached the northern Leeward Islands (fig. 2). After a three-day period of having near $50\,\mathrm{ms}^{-1}$ (97 kt) maximum 1-minute sustained winds, *Irma* intensified to $80\,\mathrm{ms}^{-1}$ (155 kt) in less than two days (Cangialosi et al., 2018) and maintained its intensity for 37 hours. According to Colorado State University, *Irma* was the first *TC*, globally, to persist for so long at this intensity. It generated the most Accumulated Cyclone Energy during a period of 24 hours ever recorded in the Atlantic basin. *Irma* was a category 5 *TC* on the Saffir-Simpson scale for three consecutive days,
the longest for an Atlantic hurricane since the satellite era. *Irma* made seven landfalls, four of which as a category 5 *TC*, and





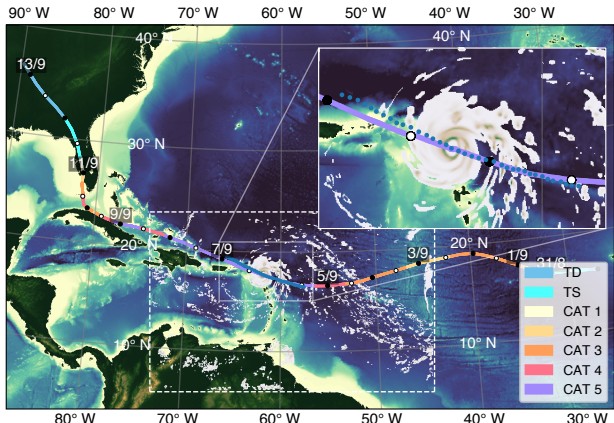

**Figure 2.** Track of hurricane Irma and modeled rain intensity. Track colour indicates Saffir-Simpson categories for *TC* intensity based on HURRDAT2. Within the model domain, modeled rain intensity at 6 September 08:00 UTC and track are shown. Best track data, HURRDAT2, and topography, ETOPO2, by National Oceanic and Atmospheric Administration (NOAA)

therefore was one of the strongest and costliest hurricanes in the Atlantic basin. The persistence of hurricane *Irma* make it a valuable subject for the isentropic vorticity budget analysis described above. A recent study on *Irma* by Torgerson et al. (2023) investigates the mechanisms during periods of rapid intensification, whereas this study examines the vorticity during a quasi-steady stage.

The remainder of this article is structured in the following order. In section 2, further details on the *PVS* flux are provided. In section 3, a brief description of the model and overview of the important methods are given. Section 4 shows the results and finally section 5 contains the conclusions of this research.

## 2    *PVS* flux

As stated previously, the vorticity flux must contain a component that counteracts the outward advection of absolute vorticity
above the atmospheric boundary layer in a *TC* environment. Therefore, a detailed description of this flux is provided below. From here on, we assume $\theta$ serves as vertical coordinate.

Under adiabatic, frictionless conditions, $P$ is conserved following the motion of an air parcel. Haynes and Mcintyre (1987) propose that $P$ can be regarded as the mixing ratio of a substance they call potential vorticity-substance (*PVS*), which is equal to $\eta$. They show in several ways that there is an exact cancellation between terms representing the cross-isentropic transport
of $\eta$, and that there are no source or sink terms within a layer bounded by two isentropic surfaces, except where the layers intersect with the surface. The results hold true in the presence of diabatic heating, frictional or other forces, regardless of the hydrostatic assumption. This implies that middle and overworld $\eta$ is simply advected like a chemical tracer and can be fully



described by a vorticity flux vector

$$\boldsymbol{J} = (u, v, 0)\sigma P + (\dot{\theta}\frac{\partial v}{\partial \theta}, -\dot{\theta}\frac{\partial u}{\partial \theta}, 0) + (\boldsymbol{F}_y, -\boldsymbol{F}_x, 0). \tag{2}$$

Here, $\dot{\theta} = \frac{d\theta}{dt}$ denotes diabatic heating, or vertical velocity in $xy\theta$-space. $\boldsymbol{F}_x, \boldsymbol{F}_y$ represent the zonal and meridional component of some external force, respectively. One may interpret the first term on the right hand side of eq. (2) as the advective vorticity flux component. The second term, involving the material diabatic heating rate, represents the effect of vertical transport of horizontal momentum. For example, in an atmosphere with positive vertical shear of the meridional wind, a localized diabatic-heating anomaly acts to transport low meridional-momentum air upward, decreasing the meridional wind in the convective region. This effectively decreases (increases) the vorticity to the west (east), and thus requires a vorticity flux in the eastward direction. Likewise, any external force, $\boldsymbol{F}$, acting on an air parcel is associated with a vorticity flux to the right of its component along an isentropic surface. Other studies have shown the usefulness of the vorticity flux in examining the response to diabatic heating in midlatitude, baroclinic disturbances (e.g. van Delden, 2003).

The vorticity flux can be separated into a slowly varying, resolved, and quickly varying, parametrized, eddy part by Reynold's decomposition as done in appendix A, resulting in

$$\begin{aligned} \boldsymbol{J} = &\left( u\eta + \dot{\theta}\frac{\partial v}{\partial \theta} - \boldsymbol{F}_y + \frac{\partial \overline{v'u'}}{\partial x} + \frac{\partial \overline{v'v'}}{\partial y} + \frac{\partial \overline{v'\dot{\theta}'}}{\partial \theta} \right)\hat{x} \\ &+ \left( v\eta - \dot{\theta}\frac{\partial u}{\partial \theta} + \boldsymbol{F}_x - \frac{\partial \overline{u'u'}}{\partial x} - \frac{\partial \overline{u'v'}}{\partial y} - \frac{\partial \overline{u'\dot{\theta}'}}{\partial \theta} \right)\hat{y}. \end{aligned} \tag{3}$$

Here, $\eta = \sigma P$ is the absolute vorticity, primed quantities indicate unresolved eddy fields, and the others resolve slowly varying fields. In the remainder of this article, any external force, $\boldsymbol{F}$, will be neglected.

The eddy velocities are estimated by Prandtl's mixing length theory $v_i' = -\boldsymbol{\ell} \cdot \nabla v_i$, with $v_i$ denoting the velocity along the $i$'th dimension and $\boldsymbol{\ell}$ a characteristic displacement. The product of any two primed quantities is proportional to $\ell^2$ and is written out explicitly in appendix A. The turbulent length scale is calculated using the procedure outlined by de Rooy et al. (2022), and multiplied by 100 to make the turbulent kinetic energy $\text{TKE} = \frac{1}{2}\ell^2\nabla^2\boldsymbol{v}$ similar to the *TKE* given by the model.

## 3 Model and methods

The model used for this study is the non-hydrostatic numerical weather prediction model *HARMONIE* (HIRLAM ALADIN Research on Mesoscale Operational NWP in Euromed) cycle 38 operated by the Royal Netherlands Meteorological Institute. It has a nominal grid resolution of 3.2 km and counts a total of 65 vertical layers with a model top at 10 hPa. The forecast is initialized and forced with a reanalysis dataset available from the ECMWF-HRES global weather prediction model and runs atmosphere-only. The version of *HARMONIE* is equipped with the *HARATU* turbulence scheme (Lenderink and Holtslag, 2004). This scheme uses a vertically integrated length scale definition that takes into account moist stability, a factor that is vital in *TC* intensification (Zhu et al., 2021). Recent improvements to this scheme have been reported by Bengtsson et al. (2017) and de Rooy et al. (2022).




*HARMONIE* has been developed mostly for weather prediction in Europe. As pointed out by Romdhani et al. (2022) on *NWP* models: their physical parametrizations might not be fully suited for a *TC* environment. Therefore, it remains important to assess whether *HARMONIE* is able to accurately predict the dynamical and thermodynamical structure of *Irma*.

The methods used to calculate various quantities in the model data are shown below.

### 3.1    Vertical interpolation

Since equation 2 is defined with potential temperature as vertical coordinate, the data provided on hybrid model levels are interpolated to isentropes. The interpolation scheme is a slightly adapted version of the procedure outlined by Edouard et al. (1997). From the top level downward, a variable $f$ within two hybrid levels $0, 1$ (increasing from top downwards) is interpolated

to some intermediate potential temperature $\theta$ level in $ln(\theta)$ coordinates using

$$f(\theta) = \frac{ln(\theta/\theta_0)}{ln(\theta_1/\theta_0)}(f_1 - f_0) + f_0). \tag{4}$$

To ensure the monotonic increase of potential temperature with height, model levels which do not satisfy a critical value $\frac{\theta_1 - \theta_0}{p_1 - p_0} \leq \frac{\partial \theta}{\partial p}_{crit} = -2 \cdot 10^{-4}$ with respect to the previous accepted level are locally rejected before the interpolation. Especially in the boundary layer, where $\theta$ often decreases with height, the critical value with respect to some level above the boundary

layer may never be reached and all data below this level are disregarded. Concerns on the validity of the impermeability of isentropic surfaces to $\eta$ in regions where $\theta$ does not increase with height (Kieu and Zhang, 2012) are deemed irrelevant when applying this level selection in the vertical interpolation procedure.

### 3.2    Centre definition

To reduce the variation of wind velocities between time steps, a lagrangian frame of reference has been adopted, where the

time-dependent *TC* centre serves as origin. This centre is determined by finding the grid point with the least azimuthal sea level pressure variance around it in a square of 21x21 points around the minimum sea level pressure. The sea level pressure variance is calculated in a square of 41x41 grid points centered at each candidate point. This square is divided into 8 equal right-angled triangular sections whose sides align with the diagonals and perpendicular bisectors of the enclosing square. The variance is taken over the section-mean sea level pressures. For the sake of simplicity, we neglect any accelerations due to track curvature.

All azimuthal mean quantities shown in this paper are derived using three-dimensional variables, and thus provide an azimuthal mean view of the non-axisymmetric model results. Besides the lagrangrian framework, all computations have been performed in a eulerian reference frame as well, yielding very similar results in an azimuthal mean sense.

### 3.3    Diabatic heating

The second term of equation 2 depends on the diabatic heating rate, which is not directly available in the dataset. It is estimated

by applying the continuity equation in isentropic coordinates. The mass flux $I = (I_x, I_y, I_\theta)$ can be evaluated by assuming hydrostatic balance, which is a reasonable approximation given that hydrostatically unstable and near-neutral layers have been filtered out. The total mass in a grid cell (in $xy\theta$-space) confined between adjacent isentropic surfaces $k, k-1$ (index $k$





increasing with height) is then equal to $-\frac{\Delta p}{g}\Delta x \Delta y$, where $g$ is the gravitational constant, $\Delta p$ the vertical pressure difference between level $k$ and $k-1$ and $\Delta x$, ($\Delta y$) the zonal (meridional) grid size. The horizontal parts of the mass flux are now simply

obtained by multiplying the mass with the average zonal and meridional velocities between layers $k, k-1$, and dividing by volume. The vertical differences in the cross-isentropic mass flux are now calculated by evaluating the mass tendency in a grid cell and subtracting the mass flux divergences integrated over the grid-enclosing area. The specific mass continuity equation for evaluating these vertical differences is given by

$$(I_\theta)_{k-1} - (I_\theta)_k = \frac{-1}{g}\left(\frac{\partial(\Delta p)}{\partial t} + \frac{\partial(\Delta p \Delta y(u_k + u_{k-1}))}{2\Delta y \partial x} \right.$$
$$\left. + \frac{\partial(\Delta p \Delta x(v_k + v_{k-1}))}{2\Delta x \partial y}\right), \tag{5}$$

These differences can be integrated from the highest isentrope downward to obtain the cross-isentropic mass flux, $I_\theta$. For the lowest isentropic level above ground, $\Delta p$ is taken with respect to surface pressure and $(u,v)_{k-1}$ are replaced by the corresponding 10 m velocities. The diabatic heating rate is related to the cross-isentropic mass flux by $\dot{\theta} = I_\theta \sigma^{-1}$ where $\sigma = \frac{-1}{g}\frac{\Delta p}{\Delta \theta}$ is the isentropic density between the two layers $k, k-1$ and $\Delta \theta$ the vertical potential temperature difference.

### 3.4   Flux integration

The tendency of volume-integrated isentropic absolute vorticity can be expressed by the closed area integral over the control volume of the horizontal vorticity flux, $\boldsymbol{J}$ (van Delden, 2003, eq. 10). This flux has no cross-isentropic component, hence the vorticity anomaly can only be created by horizontal convergence of $J$. Choosing a cylindrical control volume, $V$, concentric with the *TC* axis, the corresponding tendency equation, neglecting eddy terms, is shown below.

$$\frac{d}{dt}\int_V \eta \, dV = -\oiint_A J_r \, dA$$
$$J_r = v_r \eta + \dot{\theta}\frac{\partial v_t}{\partial \theta} \tag{6}$$

Here, subscripts $r$ and $t$ denote radial and tangential components, respectively. Note that only the radial vorticity flux component matters due to the choice of control volume and net zero cross-isentropic vorticity flux. In differential form, the above equation may be written as

$$\frac{\partial \eta}{\partial t} = -\nabla \cdot \boldsymbol{J}, \tag{7}$$

where $\boldsymbol{J}$ is given by eq. 3.

Generally, the vorticity flux due to advection in a *TC* is divergent above the boundary layer, providing a mechanism for spin-down. As shown in the results, this vorticity flux component is compensated by an equal-magnitude opposite vorticity flux in the radially inward direction due to diabatic heating, prohibiting spin-down of the simulated *TC* in the free troposphere and thus ensuring the maintenance of an approximate steady-state primary circulation.

     Radial and tangential components of vector fields may be shown in the results. These are obtained by decomposing the zonal

and meridional components using the angle with respect to the *TC* centre.





## 4 Results

Since the model biases in simulating *TCs* are *a priori* unknown, the results start with a validation of the model results.

### 4.1 Validation

To evaluate the validity of the model run, the pressure and velocity fields are compared to aircraft observations. These observa-
tions were made from 05-09-2017 17:43 till 06-09-2017 02:51 UTC on board of one of the Lockheed WP-3D Orion *Hurricane
Hunter* aircraft operated by the National Oceanic and Atmospheric Administration (NOAA). The aircraft made four passages
through the eye of *Irma* at constant velocity and pressure altitude. The first of these passages was at 700 hPa, the latter three
at 750 hPa. Fig. 3 shows the extrapolated sea level pressure from this flight, as well as the calculated wind speed. The radial
pressure profile of *HARMONIE* is in close agreement with observations in the eyewall and beyond. Only within 25 km from
the centre the pressure is overestimated. The central sea level pressure is overestimated by more than 25 hPa. From the wind
profile we can see that *HARMONIE* underestimates the wind in the inner core region up to the outside of the eyewall. The max-
imum wind is therefore underestimated as well. Despite not truly accurately predicting the dynamical structure of the cyclone,
the model does produce a realistic profile that is quite similar to the observed one. The inaccuracy in sea level pressure might
be a result of the ECMWF model initialization, providing a dynamical structure that deviates substantially stronger from the
observations.

### 4.2 Vorticity flux

The results of the cross-isentropic mass flux calculation are shown in figure 4. By appropriately scaling the radial velocity,
a transverse velocity vector is constructed from the radial and vertical velocities. This cross section clearly demonstrates the
secondary circulation, consisting of a shallow inflow layer near the surface, cross-isentropic upwelling in the eyewall and an
outflow layer at greater height. Along the inside of the slanting upper eyewall region, there is radiative cooling due to the
absence of high cloud cover.

In fig. 5a,b the azimuthal mean radial cross sections of the advective and diabatic components of the radial vorticity flux
are shown, respectively. The advective vorticity flux in the eyewall is generally radially outward due to the tilting of constant
angular momentum surfaces. In the absence of a counteractive inward vorticity flux, this would result in a decrease of $\eta$ within
the inner eyewall region and a spin-down of the vortex. Panel b shows however that this positive adiabatic divergence of vorticity
is compensated by an approximately equal magnitude vorticity flux in the radially inward direction due to diabatic heating.
Both radial flux components show a lower maximum around (30 km, 312 K), and a higher maximum around (40 km, 355 K).
The lower maximum coincides with the location where air parcels from the inflow layer with high angular momentum have
ascended into the free atmosphere and obtained a locally maximum radial velocity due to the response to vanishing frictional
influence. This location is slightly above and radially outwards of a maximum in cross-isentropic mass flux, toward the radius
of maximum vertical wind shear. Therefore the location corresponds to a local minimum in the diabatic flux component. The
higher maximum is indicative to the transition from a mainly upward rising motion to a strongly divergent flow in the outflow

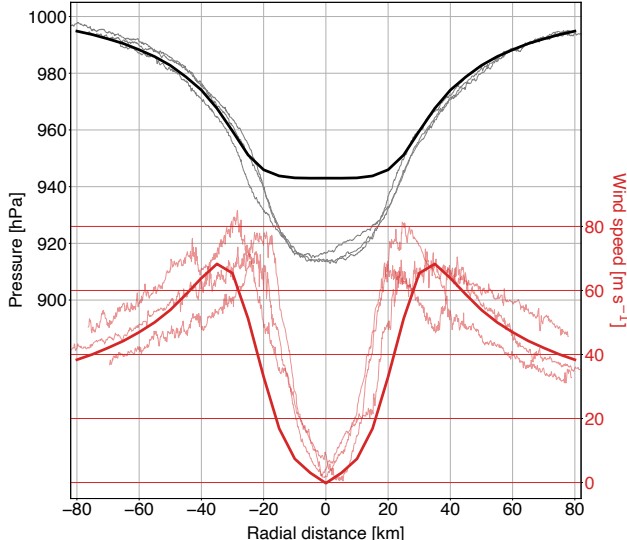

**Figure 3.** Modeled azimuthal mean sea level pressure and 750 hPa wind speed with respect to the *TC* centre (thick lines) compared to aircraft observations (thin lines). The measurements originate from a single reconaissance flight that made four passages through the center of *Irma* around 05/09 21:39, 22:45, 06/09 00:03 and 01:21 UTC. The first of these passages, excluded for consistency, was performed at 700 hPa, the latter three at 750 hPa. Sea level pressure from reconaissance data are obtained by extrapolating observed flight level pressure. The data are oriented such that the flight direction aligns with increasing radial distance, i.e. data on negative (positive) distance values are measured while approaching (leaving) the *TC* centre. The model data is averaged over the 05/09 21:00–06/09 02:00 UTC period, covering all passages.

layer. The maximum advective flux is located close to the eyewall due to the high absolute vorticity. At the same location a minimum in the diabatic flux is present due to strong vertical wind shear extending from the outflow layer downwards into the eye along the inside of the radius of maximum wind, where it meets the air continually rising at the radius of maximum wind. Figure 5c shows the divergence of the net radial vorticity flux in contours together with the average vorticity tendency during the same period. From the vorticity budget one may expect $\eta$ to increase in regions where the radial flux diverges, i.e. the radial gradient of the vorticity flux is positive. The lower maximum in advective flux is slightly stronger than the lower minimum in diabatic flux, resulting in a positive radial gradient on the inside of this maximum and a negative gradient on the outside. A similar but reversed structure may be noted near the higher maximum (fig. 6). The increase in $\eta$ calculated from the wind fields near these maxima seems to coincide with the regions of flux divergence, as expected. The magnitude of the expected $\eta$-tendency based on the flux divergence does however deviate from the modeled tendency with 4 (3) compared to 0.7 (0.5) $10^{-7}\mathrm{s}^{-2}$ being the approximate values for the high (low) maxima in *PVS* tendency. Notably the lower region of strong radial divergence at 20 km does not correspond to a diminished $\eta$-tendency, although its value is significantly lower compared to the adjacent region of strong convergence. From the results of Reynold's averaging of the advective and diabatic vorticity flux components by separating the azimuthal mean and azimuthal eddy terms, not shown in the figures, it appears that both



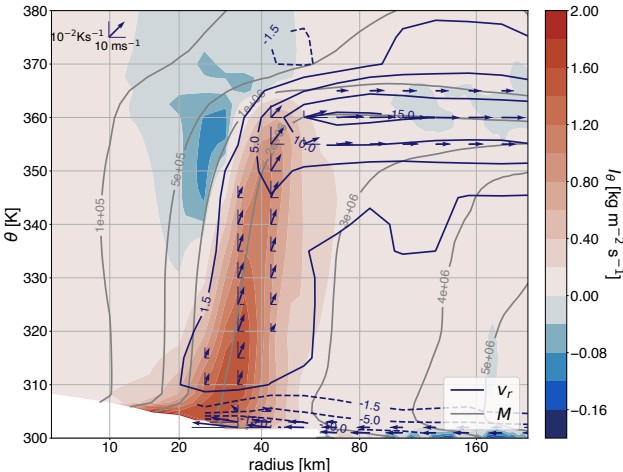

**Figure 4.** Azimuthal mean radial cross section of the cross-isentropic mass flux, absolute angular momentum per unit mass in $\mathrm{m^2 s^{-1}}$, radial velocity in $\mathrm{ms^{-1}}$ and the transverse velocity vector $10^{-3} u\hat{r} + \dot{\theta}\hat{\theta}$. The transverse velocity is shown where its magnitude exceeds 0.006. Note that the constant in the radial term is arbitrary and has been chosen for optimal display of the secondary circulation. All fields are time-averaged over the 6 September 00:00–12:00 UTC period.

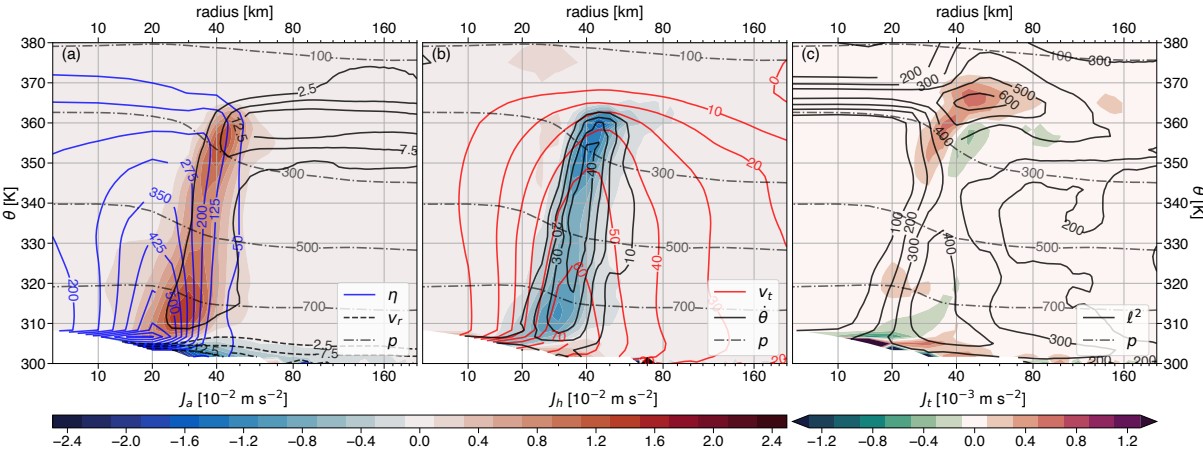

**Figure 5.** Azimuthal mean, time mean view of radial vorticity flux components. Panel **a** shows the advective flux component, absolute vorticity in $10^{-5}\ \mathrm{s^{-1}}$ and radial wind in $\mathrm{ms^{-1}}$. Panel **b** contains the diabatic flux component, tangential wind in $\mathrm{ms^{-1}}$ and diabatic heating in $\mathrm{Ks^{-1}}$. Panel **c** presents the turbulent flux component and the turbulent length scale squared in $\mathrm{m^2}$. All plots show isobars in $\mathrm{hPa}$ and contain data averaged over the 6 September 00:00–12:00 UTC period.

the advective and diabatic heating component of the vorticity flux are dominated by the azimuthal mean components. This indicates that *Irma*, in her quasi-steady-state, maintains its vorticity structure without eddy activity playing a significant role.





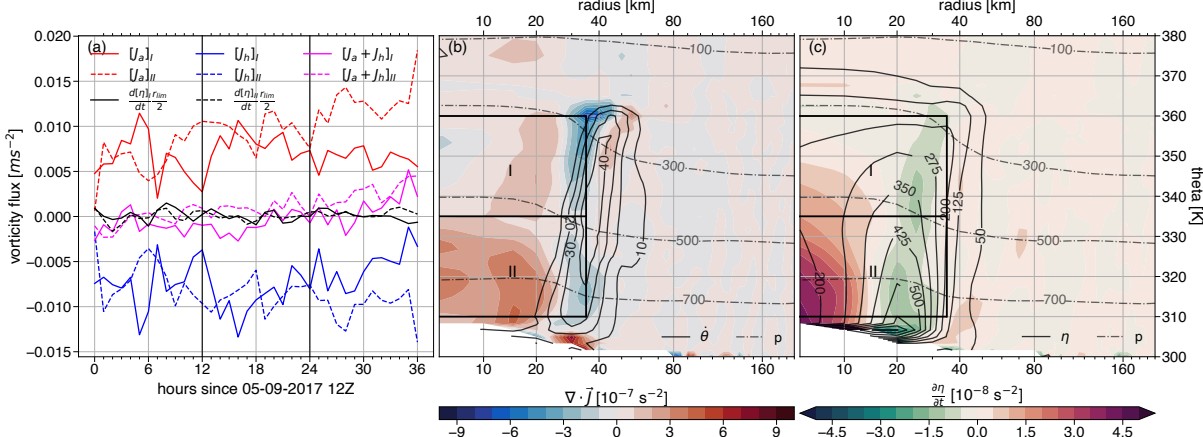

**Figure 6.** Averaged vorticity flux along boundary of control volumes and vertical cross sections of the vorticity flux divergence and $\eta$-tendency. Panel **a** shows the averages of the advective, diabatic and advective + diabatic components of the radial vorticity flux in red, blue and magenta, respectively. Each line represents the thickness-weighted average along the lateral boundary of control volume $I$ (dashed) or $II$ (solid), which are depicted in panels **b** and **c**. The black lines in panel **a** represent the volume-averaged $\eta$-tendencies multiplied by $\frac{r}{2}$, indicating the expected average vorticity flux for maintenance of the balance each control volume. Panel **b** shows a vertical azimuthal mean cross section of radial vorticity flux divergence (shading), diabatic heating (contour lines) and the cylindrical areas indicated as control volume $I$ and $II$, ranging from 310–335 K and 335–360 K, respectively. Their common lateral boundary resides at $r = 35$ km. Panel **c** shows the vertical azimuthal mean cross section of $\eta$-tendency (shading), $\eta$ (contour lines) and the control volumes. The vertical cross sections in panel **a** and **b** show time-averaged fields over the 6 September 00–12 UTC period.

In fig. 6, two cylindrical control volumes are defined extending to a fixed radius. This radius has been chosen inside the eye
230 wall because the radial vorticity flux is significant in this region. Since there is no cross-isentropic flux of vorticity, the only flux
component considered is the radial flux across the lateral boundaries. Fig. 6a shows that the advective and diabatic components
of the radial vorticity flux are approximately equal but opposite in sign. The turbulent component is left out of panel **a** as it
does not have a significant impact on the figure. The Pearson's correlation coefficient for the advective and radial components
for region $I$ ($II$) is equal to 0.88 (0.80). The sum of these two components is close to zero, considering its temporal variation.
235 The diabatic component of the vorticity flux does provide the inward vorticity flux needed to maintain a steady state. However,
there seems to be a small uptrend in the sum of the advective and diabatic components, while the modeled absolute vorticity
tendency does not trend. Perhaps, this indicates that turbulence is insufficiently parametrized as the effect of turbulence in other
cases is known to diffuse $\eta$ out of local-maximum regions (e.g. Haynes and Mcintyre, 1987). Panels **b** and **c** show the vorticity
flux divergence and $\eta$-tendency, which should be equal to one another according to *PVS* theory.



# 5 Conclusions

The extreme persistence of *TC Irma* has created a valuable case for assessing the budget of absolute vorticity in a modern *NWP* model. We have shown that during *Irma*'s lifetime peak intensity stage, the radially outward advection of absolute vorticity in the free troposphere is compensated by a radially inward vorticity flux due to diabatic heating. In the eyewall region, this counteracting flux has a strong correlation with the advective component during the whole simulation. This is a consequence of the *PVS* theorem relating the absolute vorticity in a control volume to the horizontal vorticity flux integrated across its boundary. The effect of turbulent transport is surprisingly small. A complete closure of the budget is not possible in this simulation, which might in part be due to the lack of turbulent diffusion. Other reasonable explanations for the difference include the usage of isentropic coordinates in an environment with hydrostatically instable regions, or the large time step posing a challenge for calculating tendencies. Nevertheless, the strong agreement between the vorticity flux components indicates its ability to help understand mechanisms driving *TC* intensity.

*Code availability.* The source code is available on https://github.com/JdeJong96/irma-git.git

**Appendix A: Derivation**

We start with the horizontal components of the momentum equation (A1) on an $f$-plane, neglecting curvature terms and the continuity equation (A2) in isentropic coordinates $(x, y, \theta)$:





$$\frac{Du}{Dt} = -\frac{\partial \Psi}{\partial x} + fv + F_x \tag{A1a}$$

$$\frac{Dv}{Dt} = -\frac{\partial \Psi}{\partial y} - fu + F_y \tag{A1b}$$

$$\frac{D\theta}{Dt} = \frac{\theta}{\Pi} \tag{A1c}$$

$$\frac{\partial \sigma}{\partial t} = -\boldsymbol{\nabla} \cdot \boldsymbol{I} \tag{A2a}$$

$$\boldsymbol{\nabla} = \left( \frac{\partial}{\partial x}, \frac{\partial}{\partial y}, \frac{\partial}{\partial \theta} \right) \tag{A2b}$$

$$\boldsymbol{I} = \sigma \left( u, v, \frac{D\theta}{Dt} \right) \tag{A2c}$$

where $\frac{D}{Dt} = \frac{\partial}{\partial t} + u\frac{\partial}{\partial x} + v\frac{\partial}{\partial y} + \frac{D\theta}{Dt}\frac{\partial}{\partial \theta}$ is the material derivative expressed in the current coordinate system, $u$ and $v$ are the zonal and meridional wind, $\Psi = c_p T + gz$ is the Montgomery streamfunction, using the specific heat of dry air at constant pressure $c_p$, temperature $T$, the acceleration due to gravity $g$ and the geopotential height $z$. $f$ is the Coriolis parameter, $F_x$ and $F_y$ are the unspecified frictional force components acting in the zonal and meridional direction, $\Pi = \frac{c_p T}{\theta}$ is the Exner function, $\sigma = -\frac{1}{g}\frac{\partial p}{\partial \theta}$ is the isentropic density, $\boldsymbol{\nabla}$ the del operator and $\boldsymbol{I}$ the mass flux.

To account for turbulent motion that is not resolved explicitly we now separate a rapidly fluctuating eddy component from the slowly varying (resolvable) mean flow, e.g. $u = \overline{u} + u'$ where $\overline{u}$ ($u'$) is the mean (eddy) component. By definition, the time mean of the eddy component is zero. Re-evaluating the material derivatives and dropping the averaging bars gives

$$\frac{Du}{Dt} = -\frac{\partial \Psi}{\partial x} + fv + F_x - \frac{\partial u'u'}{\partial x} - \frac{\partial u'v'}{\partial y} - \frac{\partial u'\dot{\theta}'}{\partial \theta} \tag{A3a}$$

$$\frac{Dv}{Dt} = -\frac{\partial \Psi}{\partial y} - fu + F_y - \frac{\partial v'u'}{\partial x} - \frac{\partial v'v'}{\partial y} - \frac{\partial v'\dot{\theta}'}{\partial \theta} \tag{A3b}$$

$$\frac{D\theta}{Dt} = \frac{\theta}{\Pi} - \frac{\partial \dot{\theta}'u'}{\partial x} - \frac{\partial \dot{\theta}'v'}{\partial y} - \frac{\partial \dot{\theta}'\dot{\theta}'}{\partial \theta} \tag{A3c}$$

Combining the horizontal momentum equations by taking the y-derivative from the x-momentum equation and the x-derivative from the y-momentum equation and subtracting the former from the latter we obtain:

$$\begin{aligned}
&\frac{D(\zeta + f)}{Dt} + (\zeta + f)\delta + \frac{\partial v}{\partial \theta}\frac{\partial \dot{\theta}}{\partial x} - \frac{\partial u}{\partial \theta}\frac{\partial \dot{\theta}}{\partial y} = \frac{\partial F_y}{\partial x} - \frac{\partial F_x}{\partial y} \\
&+ \frac{\partial^2 u'u'}{\partial x \partial y} + \frac{\partial^2 u'v'}{\partial y^2} + \frac{\partial^2 u'\dot{\theta}'}{\partial y \partial \theta} - \frac{\partial^2 v'u'}{\partial x^2} - \frac{\partial^2 v'v'}{\partial x \partial y} - \frac{\partial^2 v'\dot{\theta}'}{\partial x \partial \theta}
\end{aligned} \tag{A4}$$

where



$$\zeta = \frac{\partial v}{\partial x} - \frac{\partial u}{\partial y} \tag{A4a}$$

$$\delta = \frac{\partial u}{\partial x} + \frac{\partial v}{\partial y} \tag{A4b}$$

Defining $\eta = \zeta + f$ and rewriting the equation in flux form we obtain

$$
\begin{aligned}
\frac{\partial \eta}{\partial t} = & -u\frac{\partial \eta}{\partial x} - v\frac{\partial \eta}{\partial y} - \dot{\theta}\frac{\partial \eta}{\partial \theta} - \eta\delta \\
& + \frac{\partial u}{\partial \theta}\frac{\partial \dot{\theta}}{\partial y} - \frac{\partial v}{\partial \theta}\frac{\partial \dot{\theta}}{\partial x} + \frac{\partial F_y}{\partial x} - \frac{\partial F_x}{\partial y} \\
& + \frac{\partial^2 u'u'}{\partial x \partial y} + \frac{\partial^2 u'v'}{\partial y^2} + \frac{\partial^2 u'\dot{\theta}'}{\partial y \partial \theta} \\
& - \frac{\partial^2 v'u'}{\partial x^2} - \frac{\partial^2 v'v'}{\partial x \partial y} - \frac{\partial^2 v'\dot{\theta}'}{\partial x \partial \theta} \\
= & -\frac{\partial}{\partial x}\left( u\eta + \dot{\theta}\frac{\partial v}{\partial \theta} - F_y + \frac{\partial v'u'}{\partial x} + \frac{\partial v'v'}{\partial y} + \frac{\partial v'\dot{\theta}'}{\partial \theta} \right) \\
& - \frac{\partial}{\partial y}\left( v\eta - \dot{\theta}\frac{\partial u}{\partial \theta} + F_x - \frac{\partial u'u'}{\partial x} - \frac{\partial u'v'}{\partial y} - \frac{\partial u'\dot{\theta}'}{\partial \theta} \right) \\
= & -\nabla \cdot \boldsymbol{J}
\end{aligned}
\tag{A5}
$$

where $\boldsymbol{J}$ is the vorticity flux vector:

$$
\begin{aligned}
\boldsymbol{J} = & \left( u\eta + \dot{\theta}\frac{\partial v}{\partial \theta} - F_y + \frac{\partial v'u'}{\partial x} + \frac{\partial v'v'}{\partial y} + \frac{\partial v'\dot{\theta}'}{\partial \theta} \right)\hat{x} \\
& + \left( v\eta - \dot{\theta}\frac{\partial u}{\partial \theta} + F_x - \frac{\partial u'u'}{\partial x} - \frac{\partial u'v'}{\partial y} - \frac{\partial u'\dot{\theta}'}{\partial \theta} \right)\hat{y}
\end{aligned}
\tag{A5a}
$$

Now the rapidly varying eddy components can be described by a local change due to some characteristic (small) displacement $\boldsymbol{l}'$:

$$
\begin{array}{lll}
u' = -\boldsymbol{l}' \cdot \nabla u & u'u' = |\boldsymbol{l}|^2 \nabla u \nabla u & v'u' = |\boldsymbol{l}|^2 \nabla v \nabla u \\
v' = -\boldsymbol{l}' \cdot \nabla v & u'v' = |\boldsymbol{l}|^2 \nabla u \nabla v & v'v' = |\boldsymbol{l}|^2 \nabla v \nabla v \\
\dot{\theta}' = -\boldsymbol{l}' \cdot \nabla \dot{\theta} & u'\dot{\theta}' = |\boldsymbol{l}|^2 \nabla u \nabla \dot{\theta} & v'\dot{\theta}' = |\boldsymbol{l}|^2 \nabla v \nabla \dot{\theta}
\end{array}
\tag{A6}
$$


where

$$\nabla = \hat{x}\frac{\partial}{\partial x} + \hat{y}\frac{\partial}{\partial y} + \hat{z}\frac{\partial \theta}{\partial z}\frac{\partial}{\partial \theta}$$



Notice how the gradient is defined in Cartesian coordinates since the turbulent length scale provided by the model is also some
distance in meters and cannot be uniquely expressed in $xy\theta$-space. Any product of two eddy components can now be expressed
as $|l|^2 \nabla u_1 \nabla u_2$, where $|l|$ is the mixing length determined by the *HARATU* turbulence scheme. From here, the calculation can
be done numerically.

*Author contributions.* JJ: data analysis, plotting, writing; MB: supervision, feedback on writing; AD: supervision, feedback on writing

*Competing interests.* There are no competing interests.

*Acknowledgements.* Funding for this research has been received from the Dutch Ministry of Education, Culture and Science (Van Meenen:
16604027). We would like to thank Sander Tijm for sharing model output with us. We also want to thank Claudia Wieners and Guus Velders
for delivering valuable feedback on the writing.



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
