# Peer review of "On the quasi-steady vorticity balance in the mature stage of hurricane *Irma (2017)"

_EGUsphere, 2023_

## Referee Comment (RC1)

**Information**:

|  |  |
|---:|---|
| Journal: | Weather and Climate Dynamics |
| Manuscript ID: | egusphere-2023-1259 |
| Title: | On the quasi-steady vorticity balance in the mature stage of hurricane Irma (2017) |
| Authors: | Jasper de Jong, Michiel L. J. Baatsen, and Aarnout J. van Delden |

**Summary**:

This study is to examine the impermeability theorem for potential vorticity substance (PVS) on isentropic surfaces during the mature (i.e., quasi-steady) stage for the storm intensity of Hurricane Irma (2017). The examination is based on the vorticity budget analysis with a numerical simulation of Hurricane Irma. Results indicated that the radially outward vorticity flux due to the divergence was mostly canceled by the radially inward vorticity flux due to the diabatic heating under the vertical shear of the tangential wind above the atmospheric boundary layer as expected in the impermeability theorem. The results also indicated a minor contribution of parameterized turbulence to the vorticity balance during the mature stage.

**General comments**:

The authors attempted to examine the theorem based on the vorticity budget analysis with the numerical simulation. The topic has a scientific interest. Particularly, the examination of the applicability of the PVS theorem to tropical cyclones (TCs) is important for a better understanding of TC dynamics because the theorem is a basic concept to understand atmospheric dynamics. The simulation could capture the maximum intensity of Irma and maintenance of the maximum intensity. However, the budget analysis from the simulation can have some issues related to accuracy and discussions.

One critical issue is the accuracy of the budget analysis. Figure 6b (6c) shows the divergence of the total vorticity flux ($\eta$-tendency). According to Eq. (7), the $\eta$-tendency will be equal to the convergence of the total vorticity flux $\mathbf{J}$ (i.e., $-\boldsymbol{\nabla} \cdot \mathbf{J}$). I'm wondering if the pattern (or sign) of the $\eta$-tendency in Fig. 6c is opposite to that of the convergence of the total flux (which can be imagined from the divergence in Figs. 6b). If Figure 6b correctly shows the divergence of the total flux, we can expect the non-minor divergence of the vorticity flux due to the parameterized turbulence with the opposite-sign pattern to

the total-flux divergence in Fig. 6b. In fact, the storm has the radius of maximum wind (RMW) at around 30 km from the center during the analysis period (Fig. 5b). According to Eq. (A6), the generation of the parameterized turbulence can be active in the inside (approximately 20-30 km from the center) of the RMW because there is strong horizontal shear of the tangential wind of the storm. The convergence of the eddy flux due to the turbulence can be negative (i.e., corresponding to a decrease in the $\eta$) within the RMW because the turbulence will work to smooth the sharp gradient of the tangential wind near the RMW. I'm concerned that it might be difficult to ignore the vorticity flux due to the parameterized turbulence if the above consideration is correct. Thus, I strongly recommend that the authors need to evaluate the eddy flux due to the parameterized turbulence in the vorticity budget. In general, to better estimate the vorticity flux due to the parameterized turbulence, we can directly use model variables which are output as external forces in the horizontal momentum equations due to parameterized turbulence.

Other issues are related to the presentation and discussion in the manuscript (Please see specific comments).

For the above reasons, I consider that the current manuscript may need major revision for the above issues. After revising, I will reconsider the recommendation.

**Specific comments**:

1. L46-47: Figure 1a is difficult to see the eyewall location because updrafts indicating the eyewall are not indicated. Please specify the radius of the eyewall in this sentence.

2. L49-50: Did the authors quantitatively confirm the radiative cooling for the warm-core extension? Previous studies indicated that adiabatic processes associated with subsidence in the eye can be a major contribution to the development of the warming in the eye (e.g., **??**).

3. Eq. (3): Please specify the meaning of $\hat{x}$ and $\hat{y}$.

4. L224-225: Please insert the equation numbers corresponding to the explanation in the end of the sentence.

5. L216-217: It seems that the sentence is not consistent with Fig. 5c. Figure 5c shows the turbulent flux component and the turbulent length scale squared. Please clarify it.

6. L217-218: (Related to general comments,) The $\eta$-tendency may increase in regions where the radial flux "converges", according to Eq. (7).

7. L218-220: According to Fig. 5c, the total flux has negative values at radii from 20 km to 30 km and 310-K potential temperature level, which means that the lower maximum in the advective flux is slightly weaker (not stronger) than that in the diabatic flux. Please clarify it.

8. L220: For the sentence "*A similar but reversed structure may be noted near the higher maximum (Fig. 6)*", please specify the panel number of Fig. 6.

9. Appendix A: Please specify what the authors derive in the title (ex., "Derivation of XX").

**Typos**:

1. L27: "models models" $\rightarrow$ "models".

2. Eq. (4): The right-hand side is not closed by the parenthesis.

3. Fig. 6: The definition of the regions $I$ and $II$ is inconsistent with the legends of Fig. 6a. According to Fig. 6a, the result of the region $I$ ($II$) is indicated by solid (dashed) line.

---

## Referee Comment (RC3)

**Review of egusphere-2023-1259: On the quasi-steady vorticity balance in the mature stage of hurricane Irma (2017)**

Using a high resolution NWP simulation, this manuscript studies the vorticity budget of a tropical cyclone (hurricane Irma, 2017) during a 12 hour period in its mature stage when the storm was relatively steady in terms of intensity. This steady state is achieved through a balance between the outward radial flux vorticity due to divergence and the inward radial flux of vorticity due to diabatic heating. The topic is potentially of interest and the underlying analysis seems sound, however I have concerns over the motivation of the study, the presentation of the work and the discussion of the results which mean I cannot recommend this for publication at the moment. My major concerns are outlined below followed by a list of more minor points.

1) While there is a paragraph in the introduction on what the paper will do (lines 39-44), this does not really say why, or what the importance of these tasks is. Why do you want to evaluate the vorticity budget? In a sense the diabatic flux component has to balance the advective component to maintain a steady state. Any turbulent contribution is likely to be small above the boundary layer and will be diffusive so is not likely to balance advection. What would we gain from confirming this? Similarly, the budget should be closed – if not then either there is a model issue, some missing term or (more likely) there is a discrepancy in the discretisation of the terms between the model and the calculated diagnostics. A clearer statement of the problem that is being tackled would increase the impact of the paper.

2) The description of the model is rather brief and lacks key details. For example, there are model results presented in section 1, but the model itself is not actually introduced until later in section 3. From the resolution I assume it is convection permitting rather than using a convection scheme, but it doesn't say anywhere. This is crucial detail for modelling TCs. Similarly there is no mention of the microphysical scheme. We are told the HARATU turbulence scheme is used, but unless you are a HARMONIE user this probably doesn't mean very much. Aspects of the turbulence scheme (i.e. the turbulent length) are used later so a bit more detail might be warranted. What length simulations are being run here? How long in are the results shown?

3) Throughout there are examples of ideas which are not clearly or thoroughly explained. This makes the paper difficult to follow. I have given some more specific examples below, but please review the manuscript from the perspective of a reader. What do they need to know to understand and interpret the results presented?

4) The conclusions are rather brief and do not really highlight the novelty in the results. A number of possible sources of error in the budget are identified, without much real though about whether these can be addressed or what their impact might be. There is no real clear indication of how this paper is a step forward in our understanding or how the results might be of use. Just presenting the budget comes across as quite a technical exercise. Using the budget to gain some new understanding / insight into the evolution of Irma for example would make for a much more interesting and novel paper. I would suggest thinking more carefully about the message you are trying to convey. This links back to point 1) above.

Minor / specific comments

Line 28. remove duplicate "models".
Figure 1. Aside from the issue that the model has not been introduced yet there is no explanation of why these pressure levels and times are used to average over. At least a line or two of justification in the text would be helpful.

Lines 57-69. This paragraph is on vortical hot towers. While I don't disagree with what you say, there is no evidence presented later that these structures are present in your simulations so it is hard to see how this is relevant without more detailed analysis.

Line 64. "usefull" → "useful"

Line 92. "middle and overworld" is a rather odd term that doesn't really make sense in English. I assume you mean that away from the boundary layer η is simply advected?

Lines 111-112. This sentence is very hard to understand without reading the de Rooy (2022) reference. Is the turbulent lengthscale the same as the displacement $l$? How is this determined? And why do you multiply by 100? What do you mean by make the TKE similar to that in the model? Isn't this scheme what the model uses? This seems entirely ad hoc. Please explain more careful.

Line 124. I agree it is important to see whether the model is able to reproduce the dynamical and thermodynamical structure of Irma, but this was not mentioned as an aim of the paper in the introduction. Perhaps mention there?

Lines 132-137. I struggled a bit to understand what you were saying here. Can you explain more clearly how you deal with cases where the potential temperature does not increase with height? Is this actually a problem? I assume it is just in the boundary layer, and you are primarily interested in η above the boundary layer? The last sentence also seemed a bit of a sweeping statement. Can you justify this a bit more?

Line 142 and surrounding lines. Using a 41x41 box around the minimum sea level pressure to find the storm centre seems a very large box (>120km). Again, I found it hard to follow the method. Perhaps include a reference if it has been used elsewhere? Or a diagram might help.

Line 146 "Lagrangian"

Line 147 "Eulerian"

Line 151-152. Is it reasonable to assume hydrostatic balance in a tropical cyclone? This seems quite a big assumption to me. I'm not sure that just ignoring the unstable regions in the model is good enough. Surely this changes the results? Do you have evidence that this is ok?

Line 194. "deviates substantially stronger" does not make sense in English. Maybe just "deviates substantially"?

Line 196-197. "By appropriately scaling the radial velocity, a transverse velocity vector is constructed from the radial and vertical velocities." I don't quite understand what you mean here? Are you just talking about the fact that you scale the velocity in the radial direction when you plot the vectors so that they can be seen clearly? If so, perhaps this could be more clearly worded to make it obvious this is just about the plotting, and not about calculating a new scaled velocity.

Line 212. "indicative to the" → "indicative of the"

Figure 3: You have shown the azimuthally averaged winds compared to in-situ aircraft observations. What is the azimuthal variability in the model? Could this explain some of the difference between the observations and model? I guess not for the pressure, but maybe for the winds? Perhaps adding the standard deviation of the model as well would be useful?

Figure 3: From the results it looks like the radius of maximum wind might be a little far out in the model. Could this be a result of the eye being poorly resolved in a 3.5km horizontal resolution model?

Lines 217-218. "… where the radial flux diverges, i.e. the radial gradient of the vorticity flux is positive." I think this statement is wrong. There is a minus sign in the equation so an increasing flux with radius (positive divergence), leads to a decrease in η.

Figure 4 caption. The caption doesn't not say what the different lines / colours / vectors are. Please include in the caption.

Figure 4 caption. Again this is confusing defining the transverse velocity vector as $10^{-3}$ u r^hat + theta^dot theta^hat. This is just scaling the components to make the vector arrows clearer on the plot?

Figure 4 caption. Why the threshold of 0.006?

Figure 6. Caption says I (dashed) and II (solid). The legend on the figure says the opposite. Which is correct?

Figure 6 caption. "The vertical cross sections in panel a and b" should be " … panel b and c"?

Line 237. "this indicates that turbulence is insufficiently parametrized". I don't understand this logic. The balance will still hold in the model whether or not the turbulence is correctly parametrized compared to reality. Since you don't actually compare with reality I'm not sure you can conclude anything about the accuracy of the turbulence parametrization from these experiments.

Line 239. I agree that the vorticity flux divergence and η-tendency should be equal, but they are an order of magnitude difference according to the scales on panels b and c. Why?

Line 247. "… might in part be due to the lack of turbulent diffusion". I don't quite follow here. There is parametrized turbulent diffusion in the model and you've shown it plays a relatively small part in the budget. Whether or not the model is accurate in terms of parametrised turbulence, the budget in the model should still close.

Line 248 "instable" → "unstable".

Eq (A6). In the middle column of equations you introduce a new variable $l$. How does this relate to $l'$? I found the notation in this equation mathematically confusing. The definition of e.g. $u'$ is fine – you take the dot product of two vectors to get a scalar. $u'u'$ is also a scalar, however you have written it as a product of the magnitude of $l$ squared (fine – this is a scalar) and two vectors (grad(u) grad(u)). This product of two vectors is not mathematically defined. I think what you actually mean is $(l'.\text{grad}(u))^2$ which is the square of a scalar and so well defined? This same incorrect notation also appears in the text on line 296. According to your definition of the eddy components you cannot simply know the mixing length, you need to know the displacement too, i.e. the vector $l$ not just $|l|$.

Appendix A: This is definitely material for an appendix as it is relatively straightfoward and much of it is "textbook". What is still missing is any explanation of how $l$ is actually derived. This is the key aspect but the reader is just referred to the model documentation at this point, after several pages of straight forward derivation of the equations. Please elaborate at little at least.

---

## Referee Comment (RC4)

Review of the revised manuscript entitled: "On the quasi-steady vorticity balance in the mature stage of hurricane Irma (2017)", by Jasper de Jong and coauthors, submitted to *EGUSphere for Discussion*
.

I am sorry to have to say that I struggled to see the point of this paper and I wonder to what readership it is aimed? In the Abstract, the authors say that "The impermeability theorem for potential vorticity substance, PVS, on isentropic surfaces provides a way to analyze the absolute vorticity structure and tendency in TCs." While this statement is uncontroversial, the question remains, what do we expect to learn from such an analysis that adds to our understanding of tropical cyclone behaviour. Put another way, what aspects of tropical cyclone behaviour do we not understand, where such an analysis can contribute to further our understanding? Reading on in the abstract, the main take-home message appears to be that "The model results agree with this theorem", referring to the impermeability theorem. In what way is this a big deal? After all, the theorem is now on par with an axiom and is always true. Its validity is incontrovertible as in `divergence of the curl of a vector field is always zero'. The proof was presented in Haynes and McIntyre 1987 J. Atmos. Sci and we would be suspicious of any study that did not adhere to the PV flux conservation law anywhere in the earth's atmosphere or oceans!

Starting at line 19 on page 1, the authors state: "This study aims to contribute to our understanding of TC intensity by analyzing the budget of vorticity in the mature, quasi-steady stage of hurricane Irma (2017)." Nowhere in the paper did I find an answer to the way in which the paper "contributes to our understanding". This understanding should be stated clearly in the text and summarized in the conclusions.

At line 39 we are told: "Additional vorticity flux components are responsible for maintaining a steady-state vortex." However, the existence of a global steady state is controversial. For example, what maintains the convection to achieve a steady state and where does the angular momentum come from to supply that lost by surface friction?

After reading the Abstract and Introduction, I read the Conclusions, where we are told that "The strong agreement between the vorticity flux components indicates its ability to help understand mechanisms driving TC intensity." How does a study of a "steady-state" hurricane indicate its ability "to help understand mechanisms driving TC intensity change". This seems non-sequitur! Moreover, how one can expect to "understand mechanisms driving TC intensity", without an analysis of the role of deep convection, which must be a major component in understanding tropical cyclone behaviour. My conclusion is that this paper needs some major rethinking.